# Chitosan Oligosaccharide Supplementation Affects Immunity Markers in Ewes and Lambs during Gestation and Lactation

**DOI:** 10.3390/ani12192609

**Published:** 2022-09-28

**Authors:** Marefa Jahan, Cara Wilson, Shawn McGrath, Nidhish Francis, Peter C. Wynn, Yuguang Du, Bruce Allworth, Bing Wang

**Affiliations:** 1Gulbali Institute, Charles Sturt University, Wagga Wagga, NSW 2678, Australia; 2School of Agricultural, Environmental and Veterinary Sciences, Charles Sturt University, Wagga Wagga, NSW 2678, Australia; 3Fred Morley Centre, Charles Sturt University, Wagga Wagga, NSW 2678, Australia; 4State Key Laboratory of Biochemical Engineering, Institute of Process Engineering, Chinese Academy of Sciences, Beijing 100190, China

**Keywords:** Chitosan oligosaccharide, ewe, lamb, immunity, body weight

## Abstract

**Simple Summary:**

Chitosan oligosaccharide (COS) is a hydrolyzate of the structural polysaccharide, chitin, extracted from crab and shrimp shells, which has significant health benefits for humans including boosting immunity. However, there is limited information on the potential use of COS for sheep health and production. We supplemented COS via a loose lick to pregnant ewes maintained on pasture for 11 weeks with an estimated COS intake @100–600 mg/d/ewe from 5 weeks pre-lambing until lamb marking at ~4 weeks of age. We demonstrated that COS is readily incorporated into sheep supplementary feed without compromising palatability. Maternal COS supplementation did not influence the body weight of ewes or lambs, but it significantly increased serum interleukin 2 concentrations in both ewes and lambs at marking and weaning, while boosting the first line of immune defence of the lamb through an increase in serum immunoglobulin M at lamb marking. Other immune markers or cytokines were not affected by maternal COS supplementation in either ewes or lambs.

**Abstract:**

Chitosan oligosaccharide (COS) is derived through deacetylation of chitin from crustacean shells. Previous studies reported the benefits of COS to gut microbiota, immunity and health of host species. In this study, 120 pregnant composite ewes were subdivided into treatment and control groups in duplicate. COS was supplemented via a loose lick to provide an estimated intake of COS @100–600 mg/d/ewe for five weeks pre-lambing until lamb marking. Body weight was recorded pre-treatment for ewes, and at lamb marking and weaning for both ewes and lambs. Serum immunity markers immunoglobulin G (IgG), immunoglobulin M (IgM), immunoglobulin A (IgA), secretory immunoglobulin A (sIgA), interleukin (IL)-2, IL10 and faecal sIgA were determined for ewes and lambs at lamb marking and weaning by enzyme-linked immunosorbent assay (ELISA). We found that COS can be incorporated in sheep feed without compromising palatability. Maternal COS supplementation did not influence the body weight of ewes or lambs. It did, however, significantly increase the concentrations of serum IL2 in ewes at marking and weaning (*p* < 0.001). In lambs, COS also significantly increased the IL2 concentration at making (*p* = 0.018) and weaning (*p* = 0.029) and serum IgM at marking (*p* < 0.001). No significant effect was observed in the concentration of any other immune marker or cytokine in either ewes or lambs. In conclusion, maternal COS supplementation significantly modulated some immunity markers in both ewes and lambs. The short duration of maternal COS supplementation and optimal seasonal conditions during the trial may explain the lack of significant body weight in ewes and lambs from the COS supplementation.

**Implications:**

Our findings indicate that COS can be incorporated in sheep feed without compromising palatability and maternal COS supplementation significantly modulated some immunity markers in both ewes and lambs, implying the functional role of COS and its use as a feed additive for improving sheep production and immunity. Findings from this study will help to develop alternative nutritional strategies to reduce the incidence of infectious diseases in sheep. COS can thus be proposed as a cost-effective functional feed targeting pregnant and lactating ewes and weaner lambs to improve sheep production.

## 1. Introduction

Perinatal mortality averages 20–23%, and post-weaning mortality 6% in lambs [1,2,3]. Lamb growth to weaning is often limited by sub-optimal nutrition of the ewe [4]. Optimum development of foetuses and newborn lambs requires transportation of adequate nutrients across the placenta and mammary gland [5]. Retardation of the growth of the mammary tissue due to malnutrition of the ewes will compromise the lactation leading to reduced colostrum and milk production [5]. Inadequate colostrum intake in turn will result in insufficient immunoglobulin transfer from ewes to the newborn lambs increasing the risk to infection after birth [6]. Maternal malnutrition during gestation can also result in poor maternal behaviour [5]. The cumulative effect of all these factors influences lamb survival. Improving ewe nutrition pre- and post-gestation will result in improvements in birth weight, energy reserve accumulation, efficient utilisation of energy reserves, and the intake of colostrum for adequate heat production [5]. Therefore, balanced nutrition of the pregnant ewes during gestation is crucial for foetal development and survival at birth. 

For the past 50 years, nutrition research for sheep producers has focussed on the delivery of key rate limiting nutrients (eg energy, protein, minerals) to production tissues, namely the skeletal musculature, mammary gland, wool follicle and the reproductive system. Chitosan oligosaccharide (COS) is a mixture of oligomers of beta-1, 4-linked D-glucosamine derived through deacetylation of chitin [7]. Chitin is isolated from the shells of crustaceans and is the second most abundant biopolymer in the world [8]. COS has many beneficial biological properties including anti-inflammatory [9,10], anti-microbial [11], anti-diabetic [12], anti-cancerous [13] and immunostimulatory properties [14]. Alam et al. [15] have reported that COS is an effective therapeutic for calf diarrhoea by effectively inhibiting the growth and pathogenicity of *S. typhimurium* and *E. coli.* COS treatment decreased the abundance of faecal *Bacteroides,* harmful bacteria *Desulfovibrio* and *Escherichia/Shigella* and increased the abundance of *Parabacteroides and Akkermansia* in mice [8]. The gut microbiota provides essential health benefits to its host, particularly by up-regulating immuno-sensitivity [16]. Moreover, the ability of COS to pass directly to the intestinal mucosa allows the beneficial probiotic bacteria to survive more readily, while the resident pathogenic bacteria suffer from the additional competition. In addition, COS has an antibacterial effect on infectious diseases, such as cryptosporidiosis. [17]. The oral administration of COS at doses of 100, 500 and 1000 mg/kg for 7 d significantly reduced oocyst excretion in experimentally induced cryptosporidiosis in lambs [17]. Recently we reported that maternal COS supplementation during gestation and lactation in gilts increased reproductive efficiency by 4%, milk production by 35% and piglet body weight by 13% [18]. Serum immunoglobulin M (IgM) and secretory immunoglobulin A (sIgA) in piglets and faecal sIgA in gilts were also significantly increased by material COS supplementation [18]. Dietary COS improves health status in broilers for safe poultry meat production [19]. However, there is limited information on the potential use of COS for sheep health and production. 

The objectives of this study were to determine whether maternal COS supplementation during pregnancy and lactation in ewes can improve immunity and body weight of both ewes and lambs. Importantly all ewes were maintained at pasture in separate groups offered a commercial trace mineral lick supplemented with or without COS. Thus, we expect to provide a simple means of dietary supplementation through a loose lick to pregnant ewes maintained at pasture to improve their health status and that of their newborn lambs. 

## 2. Materials and Methods

### 2.1. Animals

A total of 120 pregnant 3- and 4-year-old composite ewes was sourced from the Charles Sturt University (CSU) commercial flock. Pregnancy was confirmed by ultrasound (M9VET, BCF Ultrasound, Mitcham, Australia) five weeks post joining. The ewes were stratified by age and systematically allocated to a control group (n = 60) and treatment group (n = 60). Each group was further randomly divided into two replicates, with 30 ewes in each replicate. A 15 ha paddock sown with a winter grazing cereal crop (Wedgetail wheat) was subdivided into 4 sub-paddocks. Two sub-paddocks were randomly allocated to treatment paddocks and the two remaining paddocks were allocated to control paddocks. Each paddock contained an automatic water trough and two Mineral Attachment feeders (Advantage Feeders, Wendouree, Australia), which contained a commercial loose lick (Stockmins CLS, AusFarm Nutrition, Appendix A) that was provided as a supplement in addition to the grazing wheat.

### 2.2. Chitosan Oligosaccharide Supplementation

For the treatment groups, the COS supplement was mixed with the commercial loose lick to provide an estimated COS intake of 100 to 600 mg/d/ewe based on a loose lick intake of 5 to 30 g per ewe/d. The control group received the commercial loose lick with no added COS. 

Ewes were supplemented with loose lick with and without COS for a total of 11 weeks, from 5 weeks prior to the commencement of lambing on the date of 21 May, and during lambing until lamb marking (marking) on the date of 6 August, when lambs were approximately 1 to 6 weeks of age. The lambs consumed most likely negligible or no supplement in either the control or the treatment groups, given the age of the lambs and the fact that the ewes were in good body condition, on high quality feed and observed to be milking well, meaning the lambs would have been consuming almost all or all of their diet as milk. Given the range of lambing dates, the impact of maternal COS in utero extended from 5 weeks to 10 weeks across the treated lambs. The average ewe intake of loose lick was monitored calculated by measuring, at 2 to 3 week intervals, the weight of the left-over supplement (loose lick) and subtracting that from the total weight of supplement provided. Following lamb marking subdivisions in the 15 ha paddock were removed and all ewes and lambs were combined into one group and grazed in this paddock until lambs were weaned from their mothers 15 weeks after the lambing commenced on date.

### 2.3. Sample Collection and Body Weight Measurement

All ewes were weighed using a standard weighing crate (G02603, Gallagher, Australia) at trial commencement, at lamb marking and at weaning. At each of these events, blood and faecal samples were also collected from 15 ewes selected randomly from each sub-paddock. Blood samples (5 mL) were collected by jugular venepuncture using 18 G needles and vacutainers for serum (Pacific Laboratory Products, Australia). The blood samples were then centrifuged at 1300× *g* for 20 min at 4 °C and serum samples stored at −80 °C for further analysis. Faecal samples were collected per rectum and immediately stored at −80 °C for future analysis.

All lambs were weighed at lamb marking (1 to 6 week of age) using a lamb weighing box (G02603, Gallagher, Australia), and at weaning (10 to 15 week of age), using a standard weighing crate. At each of these events, blood (5 mL) and faecal samples were collected from 15 lambs randomly selected from each sub-paddock. The procedure for processing these sample collections was as described above for the ewe samples. Overall experimental procedure is summarised in Figure 1.

### 2.4. Immunoglobulin and Anti-Inflammatory Cytokines Assay

Serum immunity markers were analysed using enzyme-linked immunosorbent assay (ELISA) following the manufacturer’s instructions. The concentration of serum immunoglobulin G (IgG), IgM and sIgA (Catalog no. 7620 and 7630 for IgG and IgM, Alpha Diagnostic International, San Antonio, TX, USA; Catalog no. MBS738604 for sIgA, My BioSource, San Diego, CA, USA), and serum pro- and anti-inflammatory interleukin (IL) IL2 and IL10 (Cusabio, China; Catalog no. CSB-E11217sh-96T and CSB-E12817sh-96T) were analysed for all non- haemolysed serum samples for both ewes and lambs at marking and weaning. 

Faecal sample preparation for sIgA analysis was carried out following the manufacturer’s instructions of ELISA (My BioSource, USA; Catalog no. MBS738604) with slight modification. Briefly, 100 mg of faecal sample was mixed with 5 mL of wash buffer on a vortex mixer until the mixture was homogenous. One ml of the mixture was transferred into an Eppendorf tube and was centrifuged for 10 min at 10,000× *g*. The supernatant was diluted at 1:1 with wash buffer (500 uL sample + 500 μL wash buffer). An automated plate washer (EL × 50 Washer, BioTek) was used to wash the plates. A microplate spectrophotometer (SpectraMax, Bio-strategy, Doylestown, PA, USA) was used to measure the optical density at a wavelength of 450 nm. All samples were analysed in duplicate. The intra and inter coefficients of variation for the assay were calculated to ensure they were less than 5%.

### 2.5. Statistical Analysis

Two-way ANOVA was used to evaluate the effects of COS on immunity markers of ewes and lambs, and for body weight of lambs at marking and weaning. To accommodate the experimental design, group (control and treatment), time (marking and weaning) and group × time interaction were specified as fixed effects and ewe/lamb ID was used as the random effect. To compare the body weight between the treatment and control ewes from the start of the project till weaning, a general linear mixed effect regression model with repeated measures was used. We checked the necessary assumptions of equal variance in overall the range of independent variables, normality of the dependent variable and used a mixed effects model to complete the data analysis. To compare the treatment effect between ewes and lambs, three-way interaction was considered among time, treatment group and age (ewe and lamb) as well as two-way interactions and main effects. Differences were considered significant at *p* < 0.05. The relationship of faecal and serum immunity markers between ewes and lambs were analysed using Pearson correlation two tail test with significant level at *p* < 0.01. This correlation analysis was performed across all data and in separate groups (treatment, control). Values are presented as means ± SEM. The analysis was completed using IBM^®^SPSS for Windows (version 26.0) (SPSS, Inc., Chicago, IL, USA).

## 3. Results

### 3.1. COS Supplement Intake by the Ewes

The total loose lick intake between the two groups of ewes were not significantly different (Figure 2A, simple *t*-test), with ewes consuming ~20 g per day, although variation occurred during the experiment, with the loose lick intake in the treatment group being 3% higher than the control group at pre-lambing, and ~15% lower than the control group during lambing to marking (*p* > 0.05, Figure 2B). Furthermore, the loose lick intake in the treatment group was similar to the control group at 2 weeks, ~10% higher than the control at 5 weeks, and 10% to 21% lower at 8 and 11 weeks after commencement of trial, respectively (*p* > 0.05, Figure 2C).

### 3.2. Effect of COS on Body Weight of Ewes and Lambs

COS supplementation did not significantly affect either the ewe’s weight during the two time points of experiment (*p* = 0.602, df = 1, F = 0.273, Figure 3A), or the weight of lambs at marking or weaning (*p* = 0.067, df = 1, F = 3.381, Figure 3A,B).

### 3.3. Effects of COS on Serum Immunity Markers of Ewes and Lambs at Lamb Marking

Maternal COS supplementation did not significantly affect the expression level of serum IgM (*p* = 0.503, df = 9, F = 1.27), IgG (*p* = 0.223, df = 9, F = 0.276) and sIgA (*p* = 0.464, df = 15, F = 1.51) and IL10 (*p* = 0.719, df = 8, F = 0.938) in ewes at marking (Figure 4A,B); however, the cytokine IL2 (*p* = 0.000, df = 9, F = 1.19) was significantly upregulated in the treatment group compared to the control (Figure 4B). For lambs at marking, the concentration of both serum IgM (*p* = 0.000, df = 7, F = 0.457) and IL2 (*p* = 0.018, df = 9, F = 1.327) was significantly higher in the treatment group compared to the control (Figure 5A,B). The other serum immunomodulatory markers, IgG (*p* = 0.348, df = 12, F = 1.194) and sIgA (*p* = 0.813 df = 30, F = 0.457), and cytokines IL10 (*p* = 0.176, df = 12, F = 1.943) for the lambs did not significantly respond to maternal COS supplementation (*p* > 0.05 Figure 5A,B), but a positive trend in improvement was evident for serum IgG and IL10 levels (Pearson correlation two tail test, *p* > 0.01 and <0.1).

### 3.4. Effects of COS on Faecal sIgA of Ewes and Lambs at Marking

The faecal sIgA concentration in the treatment groups of ewes and lambs was ~10% and ~2% higher than the control at the marking, however the difference was not statistically significant (*p* = 0.357, df = 18, F = 0.566 for ewes and *p* = 0.913, df = 19, F = 0.422 for lambs, Figure 6).

### 3.5. Effects of COS on Serum Immunity Markers of Ewes and Lambs at Weaning

The serum IgM (*p* = 0.371, df = 15, F = 0.425), IgG (*p* = 0.656, df = 13, F = 0.236) and sIgA (*p* = 0.780, df = 29, F = 1.512) concentrations in the ewes were not significantly changed by COS supplementation at weaning (Figure 7A). However, serum cytokine IL2 concentration was significantly increased (11%) in the COS ewes compared to the control (*p* = 0.000, df = 18, F = 1.196, Figure 7B). There was no significant difference in the concentration of serum IL10 between two group’s ewes (*p* = 0.940, df = 16, F = 2.186, Figure 7B). The concentrations of serum IgM (*p* = 0.269, df = 12, F = 0.104), IgG (*p* = 0.297, df = 18, F = 0.198), sIgA (*p* = 0.615, df = 15, F = 1.532), and IL10 (*p* = 0.983, df = 19, F = 2.201) in the COS treatment group lambs were not significantly different compared to the control (Figure 8 A,B). However, at weaning, the IL2 concentration was significantly higher in the lambs of treatment group compared to the control (*p* = 0.029, df = 18, F = 1.342, Figure 8B).

### 3.6. Relationship of Immunity Markers between Ewes and Lambs

At marking, the ewes had significantly higher concentration in most of the serum immunity markers compared to the lambs (*p* < 0.05, Figure 4 and Figure 5). In contrast, the concentration of IL10 in lambs was not significantly different to that of the ewes (Figure 4B and Figure 5B). At weaning, the concentration of serum immunoglobulins and cytokines in lambs was similar to those of ewes (*p* > 0.05, Figure 7 and Figure 8), except for serum IgG, which was significantly higher in the ewes compared to the lambs (*p* < 0.001, Figure 7A and Figure 8A). Faecal sIgA were not significantly different between ewes and lambs at either time (*p* > 0.05, Figure 6).

Among all the immune markers tested, there was a positive correlation between ewes and lambs. This positive relationship was significant for serum sIgA (*p* < 0.001, r^2^ = 0.643) and IL10 (*p* = 0.000, r^2^ = 0.735) when analysis was performed across all data. Furthermore, in the separate analysis of the treatment and control groups, serum sIgA concentration (*p* = 0.000, r^2^ = 0.635) in the treatment group and serum sIgA (*p* = 0.000, r^2^ = 0.666) and IL2 (*p* = 0.000, r^2^ = 0.698) concentrations in the control group were positively correlated between the lambs and ewes.

## 4. Discussion

In this study, we investigated the potential for COS to be used as a novel feed additive to improve body weight growth and promote immunomodulatory functions in both ewes and post-natal lambs. We found that COS could be successfully incorporated in a sheep supplement without affecting intake of the supplement, suggesting COS did not compromise the palatability of feed. The supplement intake of ewes was similar between the treatment and control groups and did not induce any adverse effects in either ewes or lambs at least in this study. Studies have previously shown that COS at doses of 500, 1000 and 2000 mg/kg/d administered orally did not cause any adverse effects in rats [20] and promoted production performance of feedlot lambs receiving 136 or 272 mg chitosan/kg body weight [21]. Chitosan is the precursor of chitosan oligosaccharides. We used ~400 mg COS/ewe/d (loose lick intake ~20 g/d/ewe) as a supplement for sheep without detecting any adverse effect. Therefore, our study indicates that COS can be safely and effectively fed as a supplement when mixed with commercial feed rations. Future studies will be needed to optimise the dose and refine long term effects on sheep production.

Animal plasma cells produce immunoglobulins to defend the host against infections by identifying and neutralizing foreign objects such as bacteria and viruses [22]. The use of COS as feed supplement in poultry and swine diets has the potential to enhance immunity in animals [23]. In the present study, maternal COS intervention significantly increased the serum IgM levels in their offspring at the time of marking. This finding is consistent with our previous pig species study [18] and also, with the study reported by Wan et al. [24], in which maternal COS supplementation improved the serum IgM level in the piglets. It is well recognized that the newborn animals lack sufficient immune protection and therefore, will need passive-immunity from their mother though immunoglobulins present in the colostrum [25]. The result from this study implies that the dietary supplementation of COS to ewes during pregnancy and lactation might improve the passive immunity of lambs by increasing colostrum IgM content. This assumption is supported by previous reports in which COS supplementation to sows increased IgM content in their colostrum [14,24]. As newborn farm animals are considerably more susceptible to infection than the adults, the high serum IgM in lamb serves as the first line of defence against infections and immune regulation and immunological tolerance. Additionally, our study showed that COS supplementation for 11 weeks during late pregnancy and early lactation did not affect the concentration of serum IgG, sIgA and faecal sIgA in either ewes or lambs. These results indicate that serum IgG, sIgA and IgM have different responses to maternal COS supplementation in lambs and ewes to protect against bacterial and viral infections.

The immune modulation role of COS via material intervention may assist in protection against gastrointestinal parasites and infectious diseases. A recent study indicated that COS can have a positive effect on both clinical signs and on stool character in lambs experimentally infected with cryptosporidiosis, and can reduce oocyst excretion [17]. However, the detailed molecular mechanisms of how maternal COS supplements regulate the immune system of lambs need to be further investigated.

Cytokines are a group of soluble proteins that have a specific effect on the interactions and communications between cells and produced predominantly by helper T cells [26]. Serum cytokines level can be used to assess the innate immune system response to infection or inflammation [27]. In this study, we selected IL2 and IL10 to assess the effect of COS in the immune response of ewes and lambs, based on a bibliometric analysis dedicated to large domestic animal interleukins [28] because serum IL2 and IL10 are indicative of pro-inflammatory and anti-inflammatory responses, respectively, and are the top two interleukins reported in ovine species [28]. We found that while serum IL10 concentration was not affected by maternal COS intervention in either ewes or lambs, maternal COS supplementation significantly increased the serum IL2 concentration in lactating and weaning ewes, also in lambs at marking and weaning. In addition to being pro-inflammatory, IL2 promotes the development of T regulatory cells, which are a specialized subpopulation of T cells that act to suppress immune response, thereby maintaining homeostasis and self-tolerance [29]. Thus, the significantly increased levels of IL2 resulting from maternal COS supplementation found in this study further supports the hypothesis that COS supplementation of ewes in late pregnancy and lactation is likely to enhance immunity in the lambs.

The lack of any body weight difference in either the ewes or the lambs from the maternal COS supplementation in this study, despite advantages in immune status. This may have been due to the combination of using mature composite ewes (with already high rates of milk production) and the above average seasonal conditions and pasture production, which occurred during the experiment, and/or the relatively short period of COS supplementation. Therefore, further study will be needed to confirm our findings on the effects of COS on the bodyweight of both ewes and lambs.

Although there was not a remarkable influence of COS on the majority of immunity markers tested in this trial nor any effect on the growth rate of ewes and lamb, to our knowledge this is the first study demonstrating the level of different serum immunoglobulins and cytokines for ewes during lactation, and for lambs at lamb marking and weaning. We found that the levels of all immunoglobins, except IL10, in lambs were significantly lower than the levels expressed in ewes at marking (Figure 4 and Figure 5). However, as lambs approached weaning age, the concentrations of these serum immunoglobulins were similar to those of the ewes at weaning, with the exception of IgG where ewes displayed significantly higher concentrations than lambs (Figure 7 and Figure 8). The significant positive correlation in serum sIgA and IL10 between the ewes and lambs infers that their immune markers might be provided by ewe colostrum. Thus, the maternal COS supplementation of ewes in late pregnancy and lactation is likely to enhance immunity in the lambs. Another interesting finding is that, unlike all the other immunity markers, sIgA concentration in the faecal samples was higher in lambs compared to ewes (Figure 6), the first report of faecal sIgA concentration in young and adult sheep to our knowledge. Further study is necessary to explain the age effect on this faecal expression.

## 5. Conclusions

This study explored the potential of COS to be used as a new feed additive to improve production performance and immunomodulatory properties of both ewes and lambs. We found that COS can be incorporated into the sheep feeding ration without compromising palatability, as evidenced by the similar loose lick supplement intake between the control and COS supplemented animals. Maternal dietary COS supplementation improved serum IgM levels in lambs at marking and IL2 levels in both ewes and lambs at both marking and weaning, suggesting COS supplementation has an immunomodulatory role in sheep. Despite this finding, growth performance of both lambs and ewes was not significantly affected by maternal COS supplementation. To the best of our knowledge this is the first study documenting immunoglobulins IgG, IgM and sIgA and cytokine IL2, IL10 concentration in both ewes and lambs after maternal COS supplementation. Therefore, the data derived from this study can be an important reference point for further research in this area.

## Figures and Tables

**Figure 1 animals-12-02609-f001:**
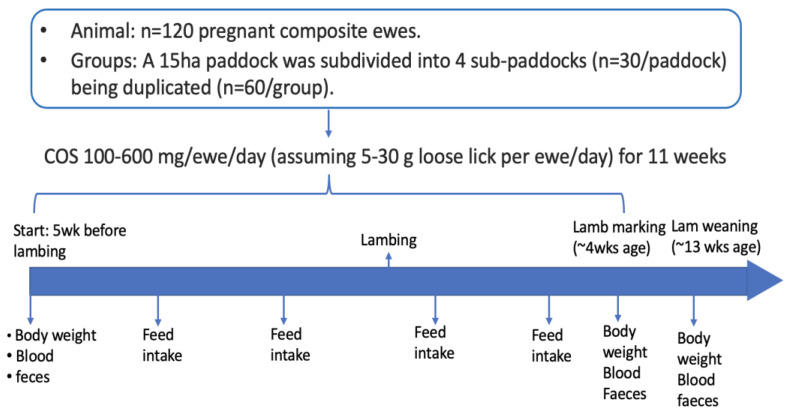
Experimental schedule of COS supplementation.

**Figure 2 animals-12-02609-f002:**
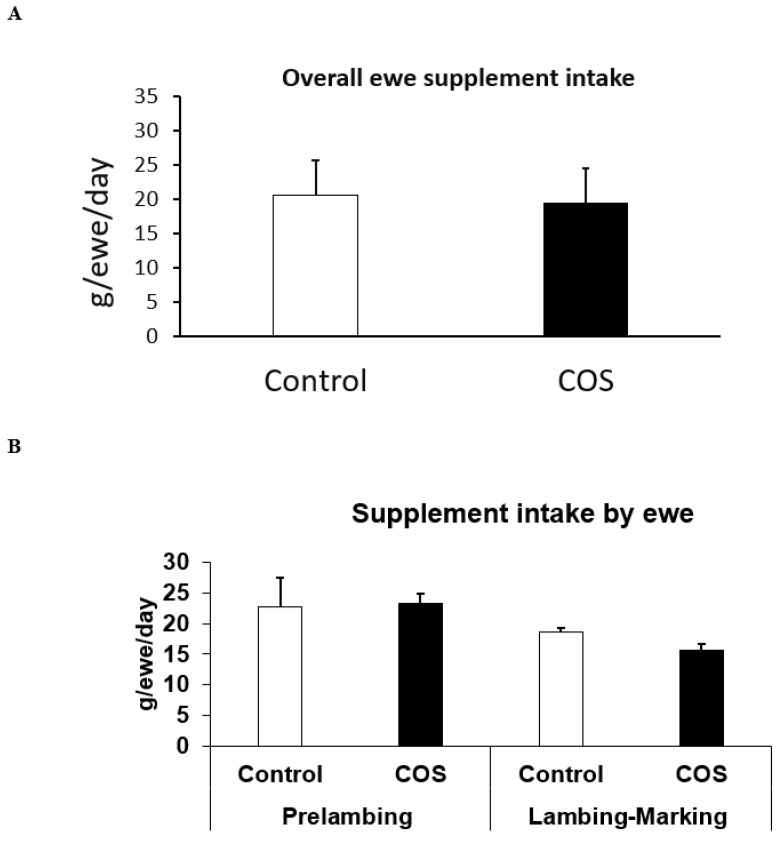
Loose lick intake between the treatment and control ewes. (**A**) Overall mean loose lick intake between the treatment and control ewes from the commencement of the trial until marking (**B**) Average loose lick intake at pre-lambing (2nd and 6th week) and lambing to marking (8th and 11th week) after the commencement of the trial (**C**). Average loose lick intake of each replicate of treatment and control group at pre-lambing 2nd and 4th week and lambing to marking 6th and 11th week after the commencement of the trial.

**Figure 3 animals-12-02609-f003:**
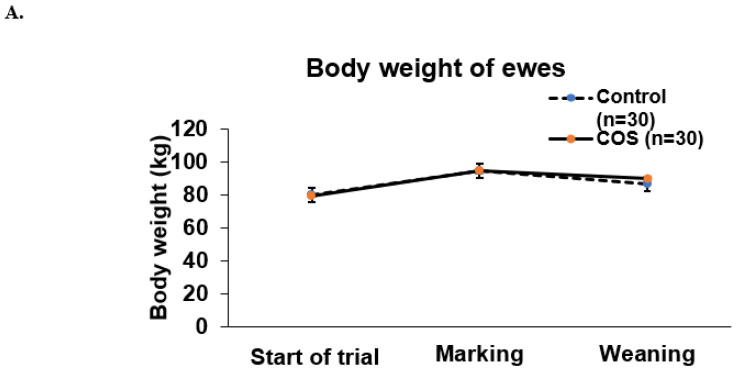
Body weight of ewes and lambs. (**A**) Mean (±standard error, SE) body weight of the treatment and control ewes from trial commencement to weaning (**B**) Mean (±SE) body weight of the treatment and control lambs measured at marking and weaning.

**Figure 4 animals-12-02609-f004:**
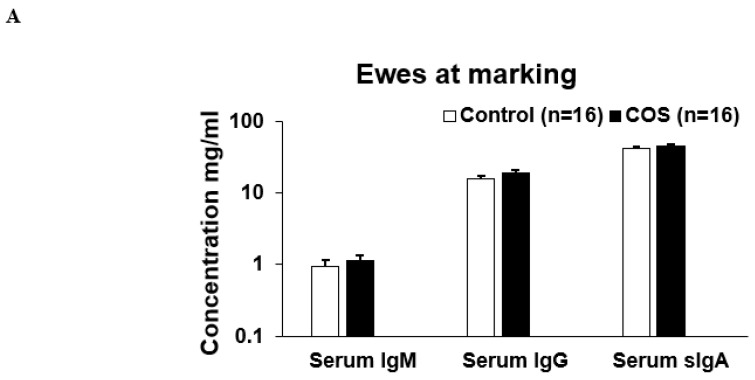
Concentration (mean ± SE) of serum immunity markers in ewes for treatment and control groups at marking. (**A**) Concentration of serum IgM, IgG, sIgA, (**B**). Concentration of serum IL (interleukin) 2 and IL10. * *p* = 0.000.

**Figure 5 animals-12-02609-f005:**
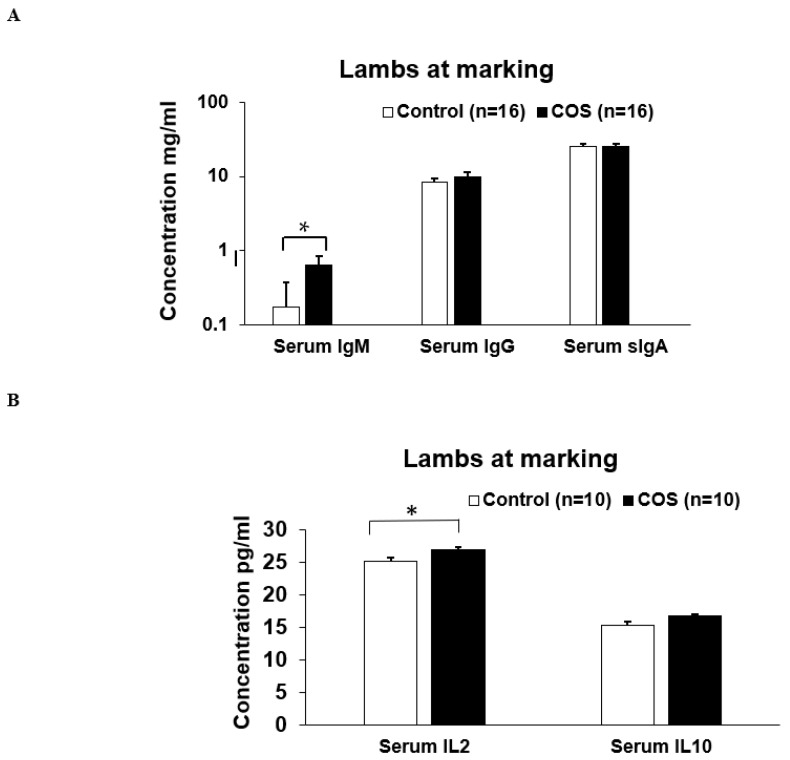
Concentration (mean ± SE) of serum immunity markers in lambs for the treatment and control groups at marking. (**A**) Concentration of IgM, IgG, sIgA (**B**) Concentration of IL (interleukin) 2 and IL10. * *p* = 0.000 and *p* = 0.018 for IgM and IL2, respectively.

**Figure 6 animals-12-02609-f006:**
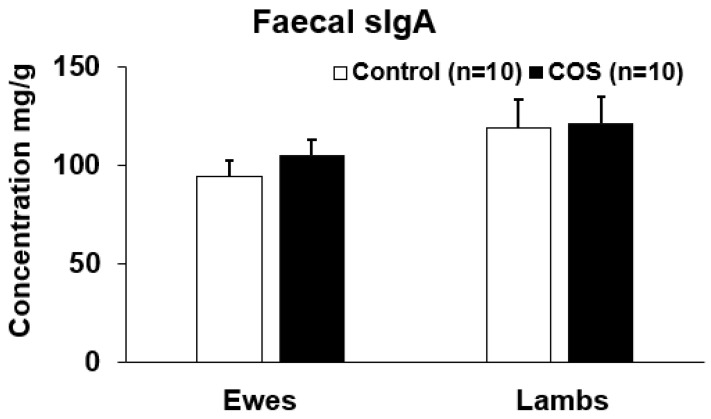
Concentration (mean ± SE) of secretory IgA (sIgA) in faeces of ewes and lambs at marking.

**Figure 7 animals-12-02609-f007:**
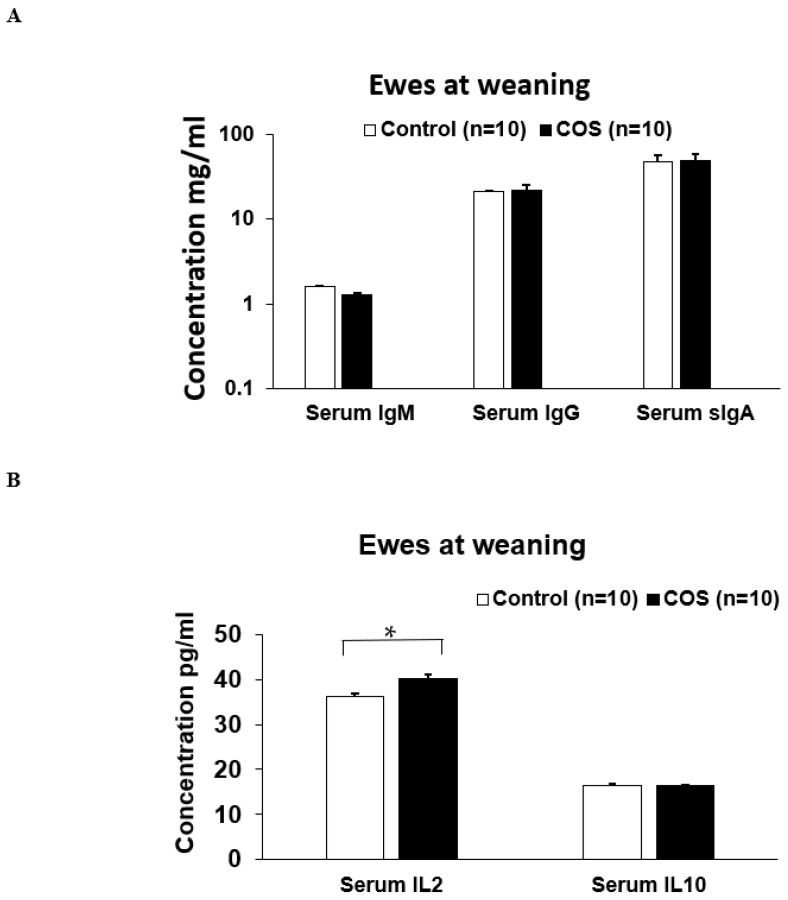
Concentration (mean ± SE) of serum immunity markers in ewes for control and treatment groups at weaning. (**A**) Concentration of IgM, IgG, sIgA (**B**) Concentration of serum IL (interleukin) 2 and IL10. * *p* = 0.000.

**Figure 8 animals-12-02609-f008:**
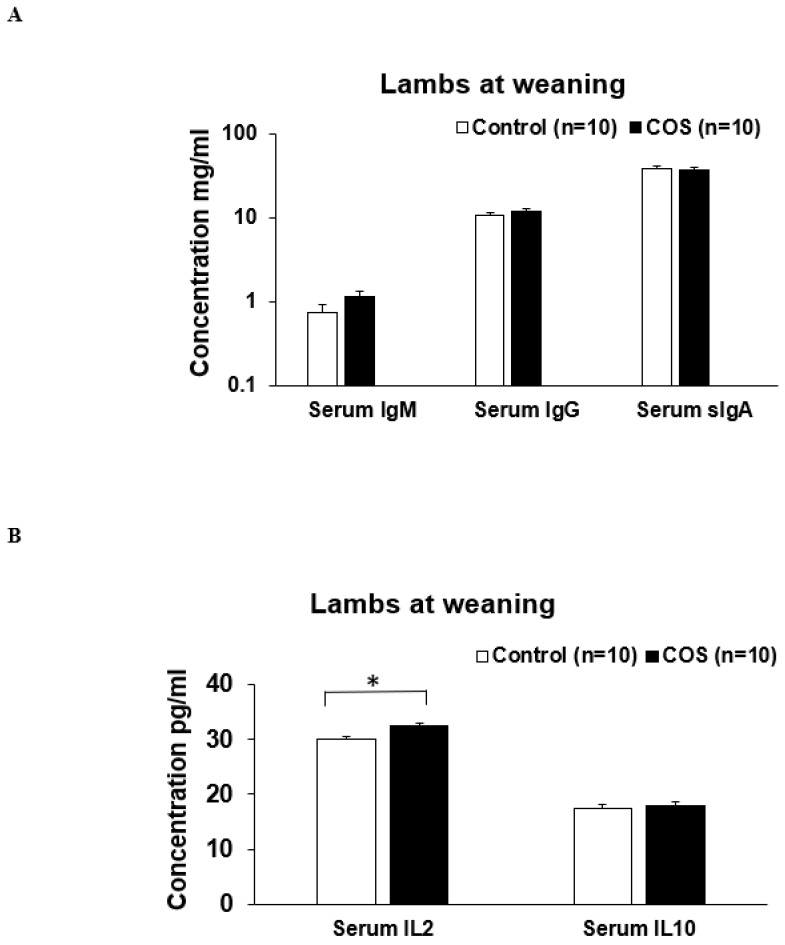
Concentrations of immunity weaning in the lamb serum samples of COS treatment and control groups of lambs collected at weaning day. (**A**) Concentration of IgM, IgG, sIgA (**B**) Concentration of IL (interleukin) 2 and IL10. Values are mean ± SE. * *p* = 0.029.

## Data Availability

None of the data were deposited in an official repository but can be available to reviewers upon request.

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
