# Peer review of "Chitosan Oligosaccharide Supplementation Affects Immunity Markers in Ewes and Lambs during Gestation and Lactation"

_animals, 2022, doi:10.3390/ani12192609_

Round 1

Reviewer 1 Report

The authors studied the impact of maternal COS supplementation during pregnancy and lactation of ewes on the immunity and live weight gain of ewes and lambs.

Comments:

1.  Why do the authors not provide SEM for the means presented in Figures 2 (A), (B), or (C)? Must be provided.   2. Verify Figure 4 B. The significance bar appears to be out of place.   3. Verify Figure 5 A. The significance bar appears to be out of place. This applies to Figure 7 B as well.   4. Line 320- Eliminate the abbreviation of immunoglobulins. You only used it once in the entire text and you already abbreviated specific immunoglobulins.   5. Figure 1 looks messy and tends to be confusing. Please edit the flow chart for the readers to better understand the experimental design.  

Author Response

Reply to reviewer 1.

We are grateful for the reviewer’s critical review of our manuscript and, importantly, their constructive comments and suggestions that have helped to improve the overall quality of our manuscript. Below, please find the reviewer's critique in bold and our response in standard text.

  1. Why do the authors not provide SEM for the means presented in Figures 2 (A), (B), or (C)? Must be provided.  

Thank you so much for your advice, yes Figures 2 A and B have been revised, and Figure 2 C is a single raw data, so there is no SEM in the figure.

  1. Verify Figure 4 B. The significance bar appears to be out of place.

Thank you, Figure 4 B has been revised, please see the revised figure in the manuscript.

  1. Verify Figure 5 A. The significance bar appears to be out of place. This applies to Figure 7 B as well.  

Thank you so much for your advice, the corrections of figure 5A and figure 7B have been made.

  1. Line 320- Eliminate the abbreviation of immunoglobulins. You only used it once in the entire text and you already abbreviated specific immunoglobulins.  

Yes, the correction has been made.

  1. Figure 1 looks messy and tends to be confusing. Please edit the flow chart for the readers to better understand the experimental design.

Thank you, please see the revised Figure 1.   

Reviewer 2 Report

This is a very interesting manuscript testing the effects of maternal chitosan oligosaccharide supplementation during pregnancy and lactation in ewes on immunity markers levels in ewes and lambs. In my opinion, it is a high value study which may contribute in several ways to the management of sheep farming. However, I believe that several improvements should take place in all sections from Introduction to Discussion and the title must be more focalized on the objectives and the findings of the study. Please see my comments and suggestions in the attached file.

Author Response

Reply to reviewer 2

We are grateful for the reviewer’s critical review of our manuscript and, importantly, their constructive comments and suggestions that have helped to improve the overall quality of our manuscript. Below, please find the reviewer's critique in bold and our response in standard text.

This is a very interesting manuscript testing the effects of maternal chitosan oligosaccharide supplementation during pregnancy and lactation in ewes on immunity markers levels in ewes and lambs. In my opinion, it is a high-value study which may contribute in several ways to the management of sheep farming. However, I believe that several improvements should take place in all sections from Introduction to Discussion and the title must be more focalized on the objectives and the findings of the study. Please see my comments and suggestions in the attached file.

Thank you so much for your encouraging positive comments. We have carefully revised manuscripts based on your suggestions. We answer all your concerns, questions, and suggestions directly in the PDF file you sent.

Round 2

Reviewer 2 Report

The revised version of the manuscript is improved indeed. However, I insist that statistical analysis has not been presented in a sufficient way. My relative comment in the original manuscript was “… you should provide the relative F-values and df. Also, the exact P values should be expressed rather than expressing inequalities (this concerns all statistical tests in the text)”

Your answer was “There is not significant difference between the groups showing at the figure clearly, nearly similar level, so it is not necessary to provide detail P value or F value to keep the MS concise.”, but this concerns only one of the many ANOVAs and GLM repeated measures you run (some of them provided significant differences). It is strongly recommended to provide F-values, df, and the exact P values for all tests, significant or not. This is a valuable information that should not be lost and will make your paper more robust.

In addition, you report in line 175 of the revised manuscript that “The necessary assumptions for this analysis were fulfilled”. Here, you should report these assumptions in more detail, i.e., what assumptions did you check to run Two-Way ANOVA and GLM with repeated measures, what statistical tests did you use for checking and provide the relative statistics.

Throughout the text you use the term gain and you performed some statistical analyses, however, in Figures as well as in the text you do not provide relative values – you just provide body weight values and not gain. This should be fixed to make your paper publishable.

Some minor comments:

Line 226: positive

Line 316: replace “showed” with “indicates”

Line 338: protect

Line 369: ewes    

Despite the fact that you inserted superscripts in several points throughout text and legends, still there are others that need correction. Please, be more careful.

Finally, I would appreciate if you answered my comments and suggestions one by one even though you do not agree with a few of them. This will make the whole process easier and less time-consuming.

Author Response

Reply to Reviewer 2

We are grateful for the reviewer’s critical review of our manuscript and, importantly, their constructive comments and suggestions that have helped to improve the overall quality of our manuscript. Below, please find the reviewer's critique in bold and our response in standard text.

The revised version of the manuscript is improved indeed. However, I insist that statistical analysis has not been presented in a sufficient way. My relative comment in the original manuscript was “… you should provide the relative F-values and df. Also, the exact P values should be expressed rather than expressing inequalities (this concerns all statistical tests in the text)”. Your answer was “There is not significant difference between the groups showing at the figure clearly, nearly similar level, so it is not necessary to provide detail P value or F value to keep the MS concise.”, but this concerns only one of the many ANOVAs and GLM repeated measures you run (some of them provided significant differences). It is strongly recommended to provide F-values, df, and the exact P values for all tests, significant or not. This is a valuable information that should not be lost and will make your paper more robust.

Point taken, we have added F value, df, and the exact P value to the revised manuscript (Please see highlight yellow).

In addition, you report in line 175 of the revised manuscript that “The necessary assumptions for this analysis were fulfilled”. Here, you should report these assumptions in more detail, i.e., what assumptions did you check to run Two-Way ANOVA and GLM with repeated measures, what statistical tests did you use for checking and provide the relative statistics.

Thank you, the necessary assumptions for this analysis were fulfilled, which means we have checked all necessary assumptions including equal variance in overall the range of independent variables, and normality of the dependent variable, because we used repeat measure (same ewe or lamb at different timepoint), repeated measures ANOVAs are particularly susceptible to violating the assumption of sphericity, therefore we use a mixed effects regression model. Please see the revised MS.

Throughout the text you use the term gain and you performed some statistical analyses, however, in Figures as well as in the text you do not provide relative values – you just provide body weight values and not gain. This should be fixed to make your paper publishable.

We use body weight to reply to body weight gain.

Some minor comments:

Line 226: positive Thanks, the correction is made.

Line 316: replace “showed” with “indicates”

Thanks, the correction is made.

Line 338: protect:

Thanks, the correction is made.

Line 369: ewes: Thanks, the correction is made.

Thanks, the correction is made, please see highlighted yellow in the revised manuscript.

Despite the fact that you inserted superscripts in several points throughout the text and legends, still there are others that need correction. Please, be more careful.

 Thank you, the correction is made, please see the revised manuscript with highlighted yellow.  

Finally, I would appreciate if you answered my comments and suggestions one by one even though you do not agree with a few of them. This will make the whole process easier and less time-consuming.

Thank you for carefully reviewing our manuscript and making all corrections.

Round 3

Reviewer 2 Report

This is a much-improved version of the original manuscript. The authors have done a good job revising the document and they have addressed all my comments. It is a high quality paper that may contribute in several ways to the management of raising sheep.